# communications
# engineering

# Coated microbubbles swim via shell buckling

Georges Chabouh [1✉], Marcel Mokbel[2], Benjamin van Elburg[3], Michel Versluis [3], Tim Segers [4], Sebastian Aland [2], Catherine Quilliet[1] & Gwennou Coupier [1✉]

Engineered microswimmers show great promise in various biomedical applications. However, their application is hindered by the slow mobility, limited maneuverability and poor bio-compatibility. Lipid coated microbubbles have high compressibility and are already approved for clinical use as diagnostic ultrasound contrast agents. Here we experimentally investigate the swimming motion of these microbubbles under external cyclic overpressure. A net displacement was generated via reproducible and non-destructive cycles of deflation and re-inflation of the microbubble. We also propose a numerical model which allows a maximum swimming speed on the order of meters per second, which falls in the range of blood flow velocity in large vessels. Unlike the acoustic radiation force technique, where the displacement is always directed along the axis of ultrasound propagation, here, the direction of propulsion is controlled in the shell reference frame. This provides a solution toward controlled steering for ultrasound molecular imaging and drug delivery.

[1] CNRS/Université Grenoble-Alpes, LIPhy UMR 5588, Grenoble F-38401, France. [2] Technische Universität Bergakademie Freiberg, Akademiestrasse, 609599 Freiberg, Germany. [3] Physics of Fluids Group, Technical Medical (TechMed) Center and MESA+ Institute for Nanotechnology, University of Twente, 7500 AE Enschede, The Netherlands. [4] BIOS/Lab-on-a-Chip Group, Max Planck Center Twente for Complex Fluid Dynamics, MESA+ Institute for Nanotechnology, University of Twente, Enschede, The Netherlands. ✉email: georges.chabouh@sorbonne-universite.fr; gwennou.coupier@univ-grenoble-alpes.fr

The creation of a programmable machine of several microns, capable of carrying out automatically a complex series of actions in our body, was the famous scientific challenge given by Richard Feynman back in 1959[1]. It was achievable only in science fiction movies (Fantastic Voyage 1966 & Inner Space 1987) until the rapid developement of microrobots research in the past two decades. Soft microrobots in particular are receiving increasing attention. Their softness, elasticity and adaptivity in their configuration is opening ways for a better interaction with the complex vascularity of the human body, compared to their rigid counterparts[2]. Softness allows for an easy shape changes, similar to what is observed in biological swimmers performing either tail or flagella beating or full deformation cycles of their body, leading to what can be called ameboid swimming[3–9], by analogy with the crawling-type ameboid motion of cells adhering on a susbstrate. However, bio-inspired artificial microswimmers share with their natural counterparts an essential flaw: they are slow. Micron-sized swimmers performing shape deformations hardly overcome the 100 µm/s limit[10–13]. Though inspiration emerging from nature has led to elegant motility strategies at microscale, this approach contains by itself intrinsic limitations set by the fluid mechanics at this scale, compromising faster displacements. This hinders their use in complex, crowded and quickly flowing media such as blood. More generally, finding the compromise between biocompatibility and swimming performance is an ongoing quest[14].

When a microrobot or a microorganism performs a shape deformation, it generates a fluid flow around it. The flow is characterized by the Reynolds number $Re = UL/v_f$, with $U$ the typical deformation velocity, $L$ the characteristic length of the microswimmer, and $v_f$ the kinematic viscosity of the surrounding fluid. In the low Reynolds number regime ("Stokes flow"), thrust (but also drag) is mediated by viscous friction with the fluid. A net displacement can only be produced if the cyclic sequence of deformations is not reversible, a principle known as the "scallop theorem"[15]. Deflation and re-inflation of a spherical shell usually involves buckling, providing the much desired feature[16] to the deformation cycle[17–21] (Fig. 1a), i.e., an hysteretic path. Furthermore, since buckling as an instability may correspond to a high Reynolds episode, it provides an additional opportunity for self propulsion, mediated this time by inertia. Swimming under such amphibolic efficiency was reported by Djellouli et al., who studied the displacement of air-filled macroscopic shells made of a rubber-like material[22] on a large range of Reynolds numbers.

Soft elastic shells also exist at the microscale. Thin shells (1−10 nm) encapsulating tiny gas bubbles (1−5 µm in radius) to avoid gas dissolution in the blood plasma are used daily in clinical practice as ultrasound contrast agents (UCAs). UCAs have shells with different visco-elastic properties depending on their material (protein, lipids or polymers)[23–25]. Coated microbubbles are highly compressible hence they present high echogenecity which is mandatory for ultrasound imaging.

The use of UCAs in the medical domain goes beyond imaging; in fact, UCAs can be used as drug carriers[26,27] whose cargo is released on-demand at the target site, thereby limiting adverse systemic effects. The release of drugs is triggered by an ultrasonic pulse localized in space and time above a threshold amplitude high enough to break the shell. Despite the huge potential of bubble-induced drug delivery systems, the circulation of the drug carriers in the bloodstream is still not controlled, which hinders their translation to clinical practice. The absence of specificity and sensitivity with the passively circulating microbubbles jeopardizes the accurate delivery of drugs.

Here, we activate UCAs by cyclic low amplitude variations of the external pressure, which is achieved in the experiments by pressurizing a closed chamber. We recently demonstrated that the anisotropy of lipid shells induces a deflation-inflation pattern, distinct from the one observed in homogeneous shell materials[21]. We examine the resulting displacements due to sequences of these non-symmetric deformations, as depicted in Fig. 1.

## Results

**Slow activation**. We first applied sinusoidal pressure cycles which varied from atmospheric pressure $P_{atm} = 101.3$ kPa to a maximum value $P_{max}$ between 110 and 160 kPa at two different frequencies $f = 1$ Hz and $f = 2$ Hz (Fig. 1). We systematically observed that when the maximal pressure $P_{max}$ was set to be slightly above a given buckling pressure $P_b$, the microbubbles would loose their spherical symmetry at threshold $P_b$ and collapse into a bowl-like shape. This deformation was reversible and could be repeated several times through slow pressure variations between $P_{atm}$ and $P_{max}$ with no apparent damage as can be seen on Fig. 1b, that shows the shape evolution within one pressure cycle.

**Ballasted microswimmer**. The fortuitous self-assembly of a three-body swimmer composed of a Sonovue microbubble ballasted by two non-buckling beads (insets in Fig. 2a, b) allows us to introduce the swimming mechanism. It was aided by the action of buoyancy that allowed for a long observation of the microswimmer over 25 cycles. The ballasted microswimmer was activated at 2 Hz with a sinusoidal pressure of amplitude 20 kPa. While the dominant motion of the swimmer, which is denser than the surrounding water, is set by its interaction with the gravity field, its trajectory shows oscillations with a frequency of 2 Hz (Fig. 2a).

A zoom on a single period reported in Fig. 2b shows that four steps can be identified, where motion is related to deformation. Upon deflation, the ballasted microswimmer undergoes a buckling event and swims up against gravity for a duration $t_1$, which results in a quasi stationary height.

After buckling, the microswimmer sediments for a duration $t_2$ while keeping a buckled shape. After that, the reinflation phase is marked by a reverse buckling event — often improperly called debuckling phase—for a duration $t_3$ during which the microswimmer sinks faster. It is followed by a slow sedimentation phase in the purely spherical shape, of duration $t_4$.

The quasi-stagnation during the buckling phase and the enhanced downward motion during the debuckling phase are strong indicators of a swimming motion that points in the center-to-buckling spot direction upon deflation and in the opposite direction upon inflation. These motions are all the more

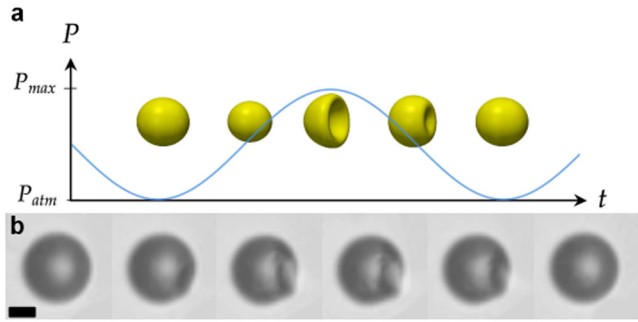

**Fig. 1 Deflation and re-inflation cycle of a coated microbubble.**
**a** Schematic of the time-asymmetric deformation cycle of an hollow shell upon an increase and decrease of external pressure $P$. **b** Snapshots of a coated microbubble, where a buckling spot appears on the top, grows then relaxes. The scale bar represents 4 µm.

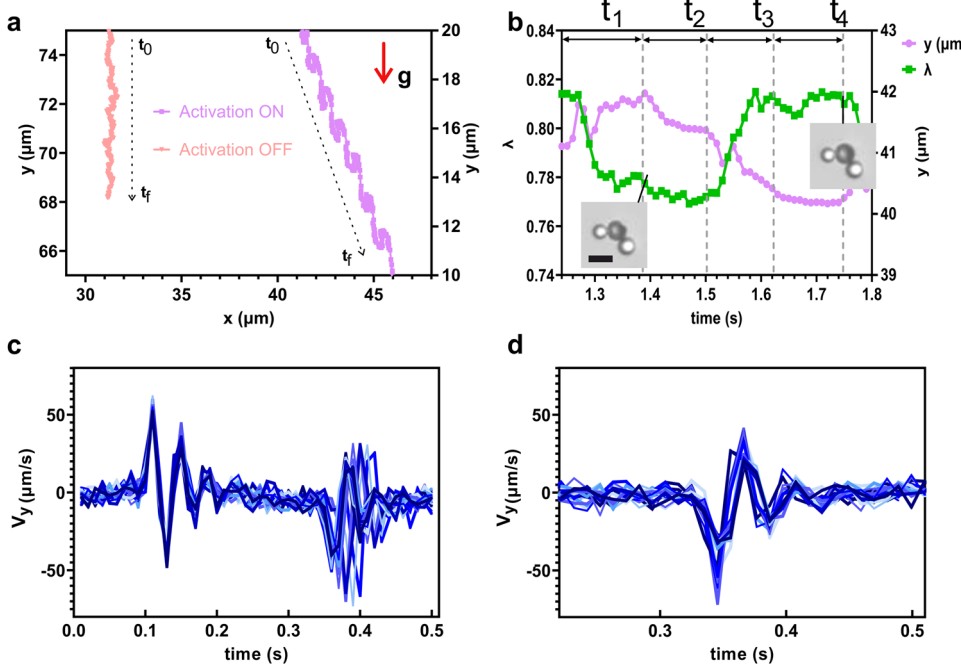

**Fig. 2 Trajectory analysis of the ballasted microswimmer. a** Part of the trajectory of the ballasted microswimmer with and without excitation (see Supplementary Video S2). The sedimentation trajectory without excitation was used to set the gravity direction $y$. **b** Zoom on the evolution of the $y$ position over one cycle (period $T = 0.5$ s), together with the evolution of the aspect ratio $\lambda$ of the shell that quantifies its deformation (see Methods section). Insets show two selected configurations, where the central shell (a UCA of radius 4.5 μm) is buckled and not buckled. $t_1$ is the duration of the buckling phase, $t_2$ of the post-buckling phase, $t_3$ of the debuckling phase and $t_4$ of the spherical phase. Vertical velocity $v_y$ (not corrected by gravity) of the ballasted microswimmer for one period $T = 0.5$ s. **c** stack of successive 25 cycles (different blue lines), **d** debuckling phase with time shift correction overlaying all 25 profiles (different blue lines). The scale bar represents 4 μm. The letter **g** denotes the gravity constant.

significant as the variations in the buoyancy during these two phases should enhance downward motion in the first case and counter-act this downward motion in the second case. We estimated the contribution of gravity (weight+buoyancy) on this microswimmer of varying volume but constant mass (see Supplementary information: Note 1) and determined that the mean net displacement in the $y$ direction $d_{net}$ is positive and bounded by: 0.16 μm/cycle $< d_{net} <$ 0.94 μm/cycle.

As far as the dynamics is concerned, both buckling and debuckling phases are marked by important oscillations of frequency slightly larger than 20 Hz (Fig. 2c). For all the 25 cycles, these oscillations start at exactly the same time for the post-buckling ones, unlike those observed upon reinflation. However, when each of the 25 profiles is individually shifted, the first peaks of these profiles overlap perfectly (Fig. 2d). The existence of a reproducible buckling threshold and of a well defined oscillation frequency illustrate the robustness of the deformation process and opens the way toward a better steering control by appropriate activation by sound waves.

**Free contrast agents**. Isolated contrast agents that float by buoyancy do not present a constant orientation $\theta_b$ (Fig. 3a) of their buckling spot, in contrast with the ballasted microswimmer; in addition to the variability between each observed microbubble, this orientation evolves in time as the microbubble rises. We showed in ref. [21] that microbubbles have a single buckling spot, which may be due to the presence of a defect or be dictated by the locus of the first buckling event. The orientation of this buckling spot may differ at each cycle because of rotational Brownian motion of the microbubbles.

Buoyancy driven rise velocity is of order 20 μm/s and varies with the volume of the microbubble: it makes it difficult to measure the component of the swimming motion in the direction

of gravity. However, the contribution $v_x$ of the swimming motion in the transverse direction is not affected by buoyancy effects. If the buckling spot has an orientation $\theta_b$ compared to $x$ axis, $v_x/\cos\theta_b$ yields, for each deformation cycle, a measure of the swimming motion in the center-to-buckling-spot direction.

The gravity direction may be potentially biased by microstreaming due to the presence of trapped air bubbles or material deformation. Therefore, in order to detect the gravity direction with more precision, we mixed the microbubble suspension with hollow glass beads (HGMS, Cospheric, USA) of 5 μm average diameter with a density of 0.1−0.7 g/cm³. These beads did not buckle when a pressure was applied and could be used as reference particles to detect the main sedimentation direction $y$.

*Correlation between deformation and motion*. The trajectory of a rising microbubble is represented in Fig. 3a. A marked deviation from the $y$ direction is visible. In this selected case where the buckling angle is constant along the cycles, $\theta_b = 150°$, it is interesting to directly quantify the correlation between deformation and motion. In Fig. 3b, the aspect ratio $\lambda$ of the bubble, averaged over 25 cycles, is plotted as a function of time, together with the velocity $v_x$ in the direction orthogonal to the main rise direction $y$. One can clearly see that upon buckling (drop in $\lambda$), the velocity $v_x$ along x peaks to a negative value with a postbuckling oscillation. Inversely, upon re-inflation, where $\lambda$ goes back to its initial value, $v_x$ peaks to a positive value. However, the absolute value of $v_x$ is smaller than upon deflation. Hence over a full cycle of deflation and re-inflation the net displacement is non-zero.

*Swimming direction and intensity*. In the above example, where $\cos(\theta_b)<0$, a net motion in the direction of the buckling pole is observed (mean $v_x$ is negative). More generally, the buckling angle $\theta_b$ can change from cycle to cycle and the information of interest is the velocity $v_b$ along the buckling axis (of angle $\theta_b$). We deduce this

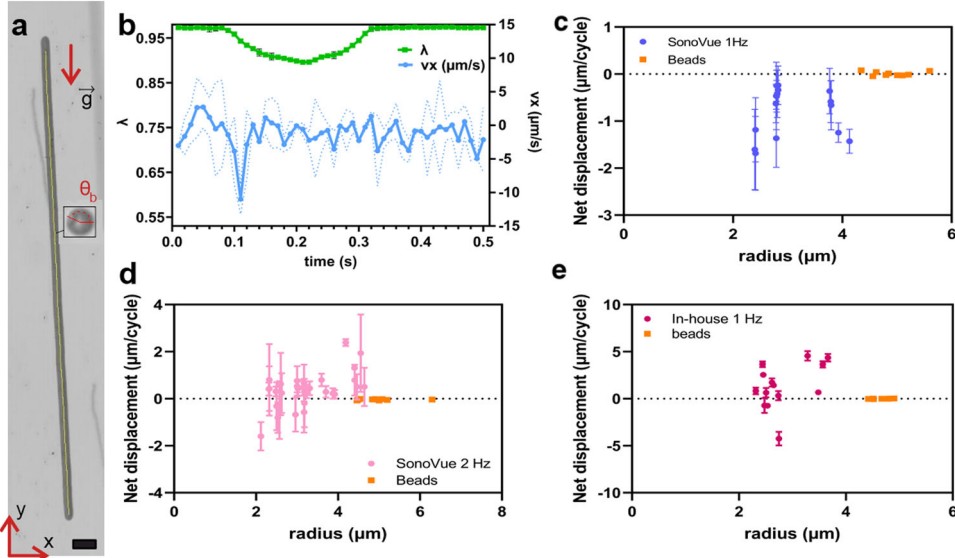

**Fig. 3 Swimming motion estimation. a** Example of microbubble track obtained by minimum intensity projection (see Supplementary Video S3), red arrow presents the direction of the corrected gravity through the reference bead, the scale bar represents 4 μm, **b** averaged aspect ratio $\lambda$ (left side) and swimming velocity $v_x$ (right side) on 25 cycles, for the same microbubble (dashed lines present the error ± standard deviation). Net displacements per cycle for SonoVue microbubbles in function of the radius at (**c**), $f = 1$ Hz, (**d**), $f = 2$ Hz, and (**e**) for in-house coated microbubbles at $f = 1$ Hz. The data shows the average net displacement on all the cycles (at least 7 cycles per shell) with the error bar being the standard deviation. The orange points indicate the net displacement per cycle (based on $v_x$) of the reference beads. The letter **g** denotes the gravity constant.

velocity from the measurement of its component along the $x$ axis: $v_b = v_x/\cos(\theta_b)$, which is calculated for each cycle. This velocity is then integrated over one cycle and averaged over all cycles, to obtain the net displacement. Figure 3c, d shows the net displacement per cycle for SonoVue microbubbles for an external driving frequency of, respectively, 1 Hz and 2 Hz. Figure 3e shows the net displacement per cycle for the microbubbles made in-house, which are stiffer than Sonovue microbubbles. The displacemens are compared to that of the non-buckling reference beads. A prominent feature in this plot is the non-zero net displacement of the microbubbles. If we consider a single point of the data, it has a high standard deviation that indicates differences in the behavior from cycle to cycle. This can result from an inaccuracy of the measurement of the buckling direction, and from the loss of the contribution in the out-of-plane direction.

In the following, we refer to a "forward" propulsion when the net displacement is positive (i.e., in the center-to-buckling-spot direction) and to a "backward" propulsion when the net displacement with respect to the buckling spot orientation is negative. For SonoVue microbubbles at $f = 1$ Hz (Fig. 3c), we notice that every coated microbubble is executing a backward motion. While for the SonoVue at a frequency of 2 Hz and the in-house microbubbles, we actually have both scenarios, forward and backward motion. Different swimming directions may be due to heterogeneities in shell properties. In fact, coated microbubbles of the same size are known to have different acoustic response[28]. The overlaid boxplot in Fig. 4a summarizes all the swimming experiments on SonoVue microbubbles for two different frequencies $f = 1$ Hz and $f = 2$ Hz and for the in-house microbubbles at one frequency $f = 1$ Hz. By comparison with the reference beads, a statistical significance in the absolute net displacement of microbubbles is shown through 2 way ANOVA test. In-house microbubbles show higher net displacement compared to SonoVue, which can be understood from their higher elastic in-plane compression modulus ($\chi = 2$ N/m for the in-house microbubbles versus ~ 0.5 N/m for SonoVue): the volume set in motion upon buckling increases with the elastic modulus[18].

## Towards tunable high velocities

While a typical motion of the micron per cycle is clearly demonstrated here, the intrinsic variability in the behavior between each coated microbubble prevents a systematic prediction for the motion of a given, unknown, UCA. Nevertheless, we can address the question of the average behavior of UCAs by means of numerical simulations. This can be useful for predicting how a colony of similar microswimmers will behave, or how a given UCA will behave after 'calibrating' the simulations, having identified the basic characteristics of its behavior.

We consider zero-thickness elastic microbubbles of initial radius $R_0$ (stress-free configuration) characterized by an in-plane compression modulus $\chi$, a Poisson ratio $\nu$ and a bending modulus $\kappa$. The characteristic distance for elasticity $d_{eff} = \sqrt{6(1+\nu)\kappa/\chi}$, equal to the thickness $d$ for a thin shell of an isotropic material, drives the deformation of shells in the buckled configuration[21].

In a recent study on the buckling threshold and final shape of UCAs, we estimated that $d_{eff}/R_0 = 10\%$[21]. The characteristic distance $d_{eff}$ would then be much larger than the typical thickness $d \simeq 5$ nm of the shell. This result, which is in line with another study on gel-phase lipid vesicles[29], is also confirmed here: by considering the thick shell, simulations show a good agreement with the observed experimental displacement. The microbubbles are immersed in a Newtonian fluid for which the Navier-Stokes equation is solved numerically. They are filled with a gas at pressure $P$ which is assumed to be instantaneously set by the shell volume according to an adiabatic process, a reasonable hypothesis considering the high velocities encountered in this problem: $PV^\kappa = P_0 V_0^\kappa$, where $V$ is the microbubble volume, $V_0$ its initial value, $P_0$ the initial pressure, and $\kappa = 1.4$. In the simulations, forward motion is always observed at slow frequency, which points to the necessity to explore the small scale to understand the backward motion observed in some experiments. Figure 4b shows that, upon slow cyclic pressurization, swimming intensity predicted by simulations is in line with experimental results, providing an effective thickness over radius ratio of 10% is chosen. This agreement leads us to confidently examine what would

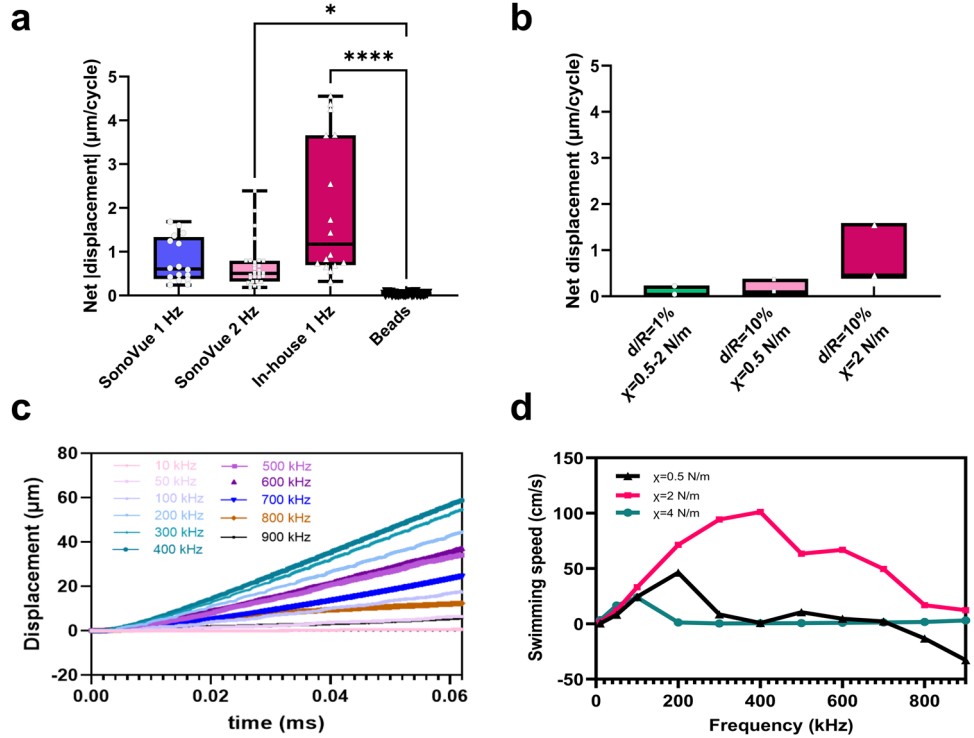

**Fig. 4 Buckling induced swimming motion through experiments and modeling. a** The absolute net displacement of SonoVue microbubbles (two different frequencies), in-house microbubbles and control (glass beads). Each symbol represents net displacement averaged on at least eight cycles. The overlaid boxplots show the median, interquartile range, mean, and the minimum to maximum values. Statistical significance of microbubbles compared to the beads. It is indicated with * ($p < 0.01$) and **** ($p < 0.0001$) using a 2 way ANOVA test on GraphPad Prism 9 software. **b** The minimum and maximum displacement per cycle from the numerical simulations at low frequency with the two assumptions $d/R$ of 1% and 10%, and initial radius in the range 1 to 5 microns. **c** Displacement per cycle in the case ($d/R = 10\%$, $\chi = 2$ N/m, $R_0 = 5$ µm) for different activation frequencies and pressure wave of amplitude 800 mbar. **d** The estimated swimming velocity in function of the varying excitation frequency for three different compression moduli $\chi = 0.5$, 2 and 4 N/m. (obtained through a linear fit of displacement ($R^2 > 0.95$), see also Fig. 5 in Supplementary information: Note 2).

be the effect of a faster activation of the coated microbubble. Figure 4c shows the displacement for sub MHz frequency activation of a shell with elastic modulus $\chi = 2$ N/m (see Supplementary Video S1). For this shell, the swimming velocity increases up to 1 m/s at $f = 400$ kHz (Fig. 4d). The quasi linear increase of the velocity with the frequency indicates a regime of independent successive buckling events. At higher frequencies, an interference with the own dynamics of the shell (whose post-buckling resonance intrinsic resonance frequency is close to 400 kHz[30]) partly inhibits the propulsion.

By variation of the shell elasticity, and for similar amplitude of the acoustic wave, the optimal swimming velocity can be reached at different activation frequencies as seen on Fig. 4d. As already explained, softer shells swim more slowly but interestingly, in the shell-wave interaction regime reached at high frequencies, they can also swim backward (Fig. 5a in Supplementary information: Note 2). More rigid shells illustrate here the subtlety of the interactions with the wave: while they swim faster at low frequency (10–50 kHz) (Fig. 5b), they are not able to buckle at higher frequencies (Fig. 5c), since they are very close to their buckling threshold (see Fig. 5d). Indeed, as pointed out by Pelekasis et al., the time needed for buckling depends both on the activating frequency and on its amplitude[31].

Thus, when considering an assembly of coated microbubbles with different elastic moduli, that are activated by a given pressure amplitude or a fixed input voltage of a medical echograph, an adequate choice of the pressure amplitude with respect to the buckling threshold of each shell appears to tune the relative contribution of each shell. In other words: one can simply control the steering of the cargo by varying the frequency.

The proposed swimming mechanism allows to reach velocities up to 1 m/s for a micron-size object performing shape deformations. This is above the performances of partially shelled microbubbles, rapidly propelled thanks to acoustic streaming[32,33], but these latter significantly influence each other[34], which impedes the realization of a multidirectionnal swimmer. Meanwhile, the possibility of "activating" only a set of coated microbubbles enables fine steering as long as the orientation of the different actors can be monitored in real time. The latter point is made possible thanks to a recent break-through in ultrasound imaging, due to in-vivo combination of ultrasound and microbubbles[35,36] in an ultrafast and super-resolved technique known as Ultrasound Localization Micro-scopy (ULM). ULM allows localization and tracking of microbubbles within 10 microns in resolution, and accurate 3D measurements of their velocities (~mm/s) deep in organs[37]. Thus, a dual-frequency ultrasound probe can achieve simulta-neously propulsion and real time imaging[38,39]. Additionally, the type of powering we studied in this article preserves the dis-placement's degrees of freedom: (i) the swimmer is activated by a scalar parameter, namely pressure, which precludes any reorientation effect that may affect, e.g., swimmers activated by magnetic fields (ii) the swimming direction is completely independent of the direction of the activation wave, by contrast with microbubble control mechanisms based on the primary radiation force ("Bjerkness forces")[40]. In a complex 3D envi-onment such as the human microvasculature where important

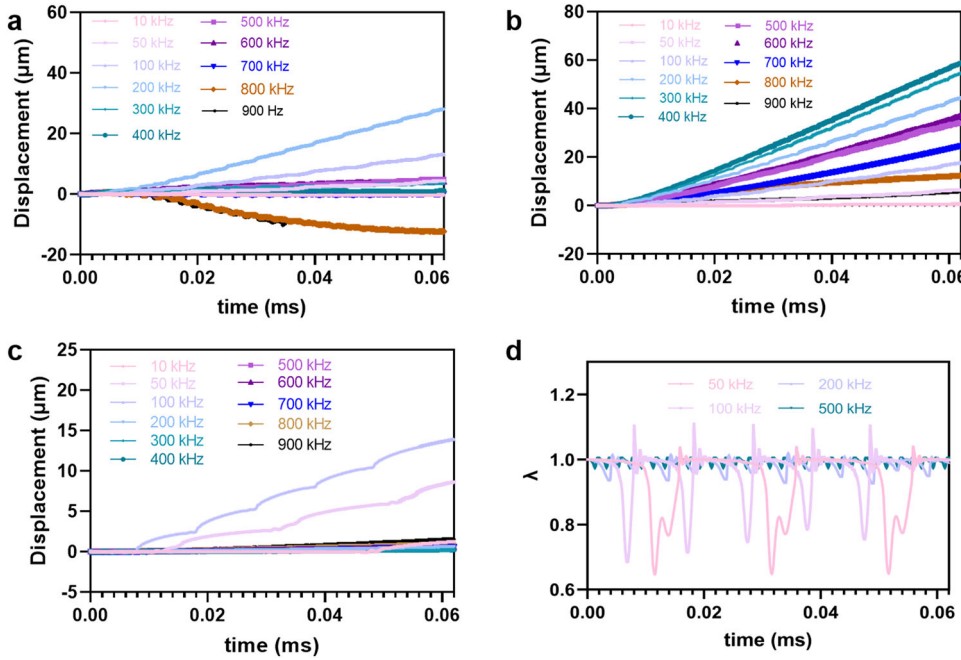

**Fig. 5 Displacement from numerical simulation. a** $\chi = 0.5\,\text{N/m}$, **b** $\chi = 2\,\text{N/m}$ and **c** $\chi = 4\,\text{N/m}$. The amplitude of pressure is set to 800 mbar and the frequency is varied. **d** Time-evolution of the aspect ratio $\lambda$ of the shell for the latter case, for some selected frequencies.

exchanges with the vessel walls take place, having a swimmer whose propelling direction is independent of the actuator features is a key point. Last but not least, a coated bubble is a robust object, biocompatible and already approved as clinical agent. Thanks to its complete coating, it has proven its ability to survive to the constraints met in the blood stream, thus making it a complete and trustful candidate for biomedical application.

UCAs were already excellent candidates for cargos enabling drug release through shell destruction and blood brain barrier opening[41]. We proved here that the same UCAs can also act as motors. Fast propulsion, detectability, and release: these are the prerequisites for a controlled microrobot dedicated to medical applications. We anticipate that these three functions can be accomplished by a compound of multiple UCAs, one as cargo and others as motors, that would all simply be activated by a single medical echograph.

## Methods
**Numerical simulations**. In the simulations, we assume that the inner pressure is initially of the order of the atmospheric pressure $P_{atm} = 101\,\text{kPa}$. We have examined coated bubbles with a radius between 1 and 5 μm and compression moduli $\chi$ in the range $0.5-2\,\text{N/m}$. The simulations are based on the numerical method presented in ref. [42], applied to the buckling of microswimmers, presented in detail and validated in ref. [30], where postbuckling oscillations were studied.

**Microbubble preparation**. SonoVue® microbubbles (Bracco Spa, Milan, Italy) consist of a phospholipid shell encapsulating a sulfur hexafluoride gas core. Before the experiment, the shells are reconstituted through a mixture of the lyophilisate with 5 ml physiological saline solution, to form a suspension that contains $\sim 2-5 \times 10^8$ microbubbles per milliliter with radius ranging from 2 to 7 μm.

We also used lipid microbubbles encapsulating $C_4F_{10}$ with a more narrow size distribution that the one we produced in a flow-focusing device. The fabrication process of these DSPC/DPPE-

PEG5000 microbubbles is summarized in ref. [21], following protocols detailed in refs. [43,44].

**Experimental methods**. As in ref. [21], the microbubble suspension was gently poured into degased water and placed in a Falcon microfluidic reservoir of 15 ml (Fisher Scientific, USA) connected to an Elveflow® pressure controller (Elvesys®, France) and to a flow-through cuvette (Aireka Scientific® Co., Ltd) using PTFE tubings.

The chamber made from quartz with a square cross section (12.5 mm × 12.5 mm) was placed under an inverted microscope (Olympus®, model IX70) which was rotated 90° through three stabilizing aluminum legs. This configuration provided an observation axis $z$ perpendicular to the gravity axis $y$. After injection in the chamber, the microbubbles rose due to buoyancy. The other end of the observation chamber was connected 1) to a valve left open to allow injection of the UCAs into the chamber through a gentle increase of the pressure in the reservoir (of order 30 mPa above atmospheric pressure), then closed for pressurization of the chamber, and 2) to a pressure sensor (MPS1, Elvesys®, France).

In order to correlate the evolution of the shell shape with the driving pressure $P_{max}$, both pressure sensor and fast camera (Miro 310, Vision Research) were triggered through the Elveflow interface. Videos were taken at a rate of 100 frame/s. An automated stage (MS-2000, ASI, USA) was used to select a microshell prior to the recording.

**Image analysis**. To track the UCAs and record their shape at each time step, we developed a tracking algorithm with sub-micron resolution[21]. The intensity profile of each image was approximated by a two-dimensional elliptic Gaussian distribution characterized by a center $(x_0, y_0)$, main axis $\sigma'_x$ and $\sigma'_y$ and orientation angle $\theta$. Data were processed so that $\sigma'_x$ would always denote the minor axis and $\theta$ its angle to the $x$ axis. The angle $\theta$ was also chosen so that when the aspect ratio $\lambda = \sigma'_x/\sigma'_y$ is minimal, the corresponding angle $\theta$ indicates the buckling direction, referred to as $\theta_b$ hereafter (i.e., a visual inspection can decide whether $\theta$ or $\theta + \pi$ should be used).

## Data availability

The data that support the plots within this paper and other findings of this study are available from the author upon reasonable request.

## Code availability

The code that support the plots within this paper and other findings of this study are available from the author upon reasonable request.

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

## Acknowledgements

G.Ch., C.Q., and G.Co. acknowledge funding from CNRS through PEPS Mécanique du futur. G.Co. acknowledges funding from Labex Tec 21 for an outgoing grant. T.S. acknowledges financial support of the Max-Planck Center Twente. Simulations were performed at the Center for Information Services and High Performance Computing (ZIH) at TU Dresden. B.vE. acknowledges financial support from NWO BUBBLE-X & ULTRA-X-TREME.

## Author contributions

G.Ch. designed the experimental protocol and set-up for buckling and swimming experiments and carried out these experiments. B.vE. produced and characterized the in-house microbubbles through acoustic attenuation of ultrasound. M.M. performed the buckling simulations. G.Ch., M.V., C.Q., and G.Co. performed the data analysis and interpretation. G.Ch., C.Q., and G.Co. conceived and designed the study. G.Ch., T.S., S.A., M.V., C.Q., and G.Co. drafted the manuscript. All authors read, corrected and approved the manuscript.

## Competing interests

The authors declare no competing interests.
