## [Peer Review File · Communications Engineering]

Reviewers' comments:

Reviewer #1 (Remarks to the Author):

The entitled 'Coated microbubbles swim via shell buckling' was carefully reviewed. In this study, experiments and numerical simulations were combined to verify that lipid coated microbubbles can generate a net displacement under external cyclic overpressure. This paper has certain innovation. However, some questions are confused.

1. Whether the microbubbles used in Fig.1 is Sonovue? The diameter of the microbubbles in Fig.1 is larger than the average diameter of Sonovue, whether it can pass through the pulmonary circulation in vivo.
2. What is the direction of given frequency? It has not been verified whether this direction has any effect on the direction of movement of the microbubbles.
3. Fig.3a does not have a scale bar. In Video 3, the degree of microbubble deformation is small and unrepresentative.
4. 'For this shell, the swimming velocity increases up to 1 m/s at $f = 400 \text{ Hz}$ '. The unit is wrong, which should be kHz.
5. How to solve the inconsistency between the imaging frequency and motility frequency of microbubbles by ultrasound?

Reviewer #2 (Remarks to the Author):

The paper presents microbubbles that can perform swimming motions through cycles of deflation and re-inflation, potentially serving as a motor for drug delivery. The work is well organized. However, the paper needs some improvements before it can be accepted.

1. The paper lacks clear descriptions of the actuation method and mechanism of the microswimmer. While it appears that the microswimmer is actuated by ultrasound, the author did not clearly state the actuation method in the manuscript. I suggest that the author add clear descriptions of the actuation method in the introduction to help readers better understand the work.
2. The author claims that the swimmer can perform swimming motion with a velocity of 1 m/s, as shown in the supplementary video. However, the scale bar and time information are not presented in the video, which makes it difficult to verify the velocity claim. I recommend that the author supplement the video with this information.
3. Since the UCAs are ultrasound contrast agents, it would be helpful to know whether they can be navigated under ultrasound imaging. The author could consider discussing this point in the paper.

4. In Figure 2, some modifications are necessary. In Figure 2b, the legend overlays the figure title. In Figure 2d, the title is incomplete. The author should make the necessary modifications to ensure that the figures are clear and easy to understand.

Reviewer #1: The entitled 'Coated microbubbles swim via shell buckling' was carefully reviewed. In this study, experiments and numerical simulations were combined to verify that lipid coated microbubbles can generate a net displacement under external cyclic overpressure. This paper has certain innovation. However, some questions are confused.

1. Whether the microbubbles used in Fig.1 is SonoVue? The diameter of the microbubbles in Fig.1 is larger than the average diameter of SonoVue, whether it can pass through the pulmonary circulation in vivo.

Thank you for your comment regarding the microbubble depicted in Fig.1. We confirm that the microbubble is SonoVue. Upon revisiting the calibration of our microscopic images, we discovered an error and have now corrected the scale bar size to 4 μm . We apologize for any confusion caused. Nonetheless, the microbubble shown in the image has an 8 μm diameter and is part of the larger fraction of the SonoVue size distribution. It represents the upper limit capable of passing through the pulmonary circulation. We specifically selected a larger bubble from the native population to aid in the microscopic visualization of the buckling phenomenon.

2. What is the direction of given frequency? It has not been verified whether this direction has any effect on the direction of movement of the microbubbles.

Thank you for your comment. In our numerical simulations, the pressure is increased uniformly in all directions without any privileged direction. Similarly, in our experimental setup, the chamber was pressurized at low frequencies of $f=1$ Hz and $f=2$ Hz, resulting in a non-directional quasi-static pressurization.

3. Fig.3a does not have a scale bar. In Video 3, the degree of microbubble deformation is small and unrepresentative.

Thank you for your comment. We have addressed your concern by adding a scale bar in the revised version of the manuscript. Regarding Video 3, we acknowledge that the 3D orientation of the coated microbubble can make it challenging to visualize its deformation. Nevertheless, the overlaid snapshot, taken from this video, in Fig. 3a demonstrates a significant deformation. Additionally, the metric used to quantify the deformation shows a representative value (see Fig. 3b). For a more detailed visualization of the deformation, we recommend referring to Video 1.

4. 'For this shell, the swimming velocity increases up to 1 m/s at $f = 400$ Hz'. The unit is wrong, which should be kHz.

We thank the reviewer for noting this typo. We have corrected the unit : kHz instead of Hz.

5. How to solve the inconsistency between the imaging frequency and motility frequency of microbubbles by ultrasound?

Thank you for this question. One can think of many ways to address this challenge: the most straightforward way is to use a dual frequency probe: one high-frequency probe for diagnostic (contrast) imaging and one low-frequency transducer for motility. Another way is to use a multiplexed 3D ultrasound probe or a sparse array where specific elements can be used to induce motility while other are dedicated to imaging.

Reviewer #2: The paper presents microbubbles that can perform swimming motions through cycles of deflation and re-inflation, potentially serving as a motor for drug delivery. The work is well organized. However, the paper needs some improvements before it can be accepted.

1. The paper lacks clear descriptions of the actuation method and mechanism of the microswimmer. While it appears that the microswimmer is actuated by ultrasound, the author did not clearly state the actuation method in the manuscript. I suggest that the author add clear descriptions of the actuation method in the introduction to help readers better understand the work.

Thank you for your advice. To clarify, in our experiments, we activate UCAs by applying cyclic overpressure to an enclosed chamber where coated microbubbles are floating. We have added a paragraph to the introduction that explains this actuation method :
Here, we activate UCAs by cyclic low amplitude variations of the external pressure, which is achieved in the experiments by pressurizing a closed chamber. We recently demonstrated that the anisotropy of lipid shells induces a deflation-inflation pattern, distinct from the one observed in homogeneous shell materials. We examine the resulting displacements due to sequences of non-symmetric deformations, as depicted in Fig.1.

2. The author claims that the swimmer can perform swimming motion with a velocity of 1 m/s, as shown in the supplementary video. However, the scale bar and time information are not presented in the video, which makes it difficult to verify the velocity claim. I recommend that the author supplement the video with this information.

Thank you for your remark and we apologize for the omission. We have updated the video with a scale bar and time information to provide more clarity on the velocity claim.

3. Since the UCAs are ultrasound contrast agents, it would be helpful to know whether they can be navigated under ultrasound imaging. The author could consider discussing this point in the paper.

We thank the reviewer for the suggestion. The frequency range used for ultrasound imaging (2-8 MHz) is higher than the navigation frequency (100-800 kHz). The most straightforward way is to use a dual frequency probe: one high-frequency probe for diagnostic (contrast) imaging and one low-frequency transducer for motility. (The following review «*Sensors* **2014**, *14*, 20825-20842; doi:10.3390/s141120825 » shows the main advances in this direction. Another way is to use a multiplexed 3D ultrasound probe or a sparse array where specific elements can be used to the motility and others for imaging.

The following paragraph has been added in the revised manuscript: Thus, a dual-frequency ultrasound probe can achieve simultaneously propulsion and real time imaging^{38,39}

4. In Figure 2, some modifications are necessary. In Figure 2b, the legend overlays the figure title. In Figure 2d, the title is incomplete. The author should make the necessary modifications to ensure that the figures are clear and easy to understand.

Thank you for bringing this to our attention. We have made the necessary modifications to these figures to ensure they are now clear and easy to understand.

REVIEWERS' COMMENTS:

Reviewer #1 (Remarks to the Author):

Authors have responded most of my comments and thus I am pleased to recommend it to be accepted for publication.

Reviewer #2 (Remarks to the Author):

The paper presents microbubbles that can perform swimming motions through cycles of deflation and re-inflation, potentially serving as a motor for drug delivery. The work is well organized. However, the paper needs some improvements before it can be accepted.

1. The paper lacks clear descriptions of the actuation method and mechanism of the microswimmer. While it appears that the microswimmer is actuated by ultrasound, the author did not clearly state the actuation method in the manuscript. I suggest that the author add clear descriptions of the actuation method in the introduction to help readers better understand the work.
2. The author claims that the swimmer can perform swimming motion with a velocity of 1 m/s, as shown in the supplementary video. However, the scale bar and time information are not presented in the video, which makes it difficult to verify the velocity claim. I recommend that the author supplement the video with this information.
3. Since the UCAs are ultrasound contrast agents, it would be helpful to know whether they can be navigated under ultrasound imaging. The author could consider discussing this point in the paper.
4. In Figure 2, some modifications are necessary. In Figure 2b, the legend overlays the figure title. In Figure 2d, the title is incomplete. The author should make the necessary modifications to ensure that the figures are clear and easy to understand.

Reviewer #2: The paper presents microbubbles that can perform swimming motions through cycles of deflation and re-inflation, potentially serving as a motor for drug delivery. The work is well organized. However, the paper needs some improvements before it can be accepted.

1. The paper lacks clear descriptions of the actuation method and mechanism of the microswimmer. While it appears that the microswimmer is actuated by ultrasound, the author did not clearly state the actuation method in the manuscript. I suggest that the author add clear descriptions of the actuation method in the introduction to help readers better understand the work.

Thank you for your advice. To clarify, in our experiments, we activate UCAs by applying cyclic overpressure to an enclosed chamber where coated microbubbles are floating. We have added a paragraph to the introduction that explains this actuation method :
Here, we activate UCAs by cyclic low amplitude variations of the external pressure, which is achieved in the experiments by pressurizing a closed chamber. We recently demonstrated that the anisotropy of lipid shells induces a deflation-inflation pattern, distinct from the one observed in homogeneous shell materials. We examine the resulting displacements due to sequences of non-symmetric deformations, as depicted in Fig.1.

2. The author claims that the swimmer can perform swimming motion with a velocity of 1 m/s, as shown in the supplementary video. However, the scale bar and time information are not presented in the video, which makes it difficult to verify the velocity claim. I recommend that the author supplement the video with this information.

Thank you for your remark and we apologize for the omission. We have updated the video with a scale bar and time information to provide more clarity on the velocity claim.

3. Since the UCAs are ultrasound contrast agents, it would be helpful to know whether they can be navigated under ultrasound imaging. The author could consider discussing this point in the paper.

We thank the reviewer for the suggestion. The frequency range used for ultrasound imaging (2-8 MHz) is higher than the navigation frequency (100-800 kHz). The most straightforward way is to use a dual frequency probe: one high-frequency probe for diagnostic (contrast) imaging and one low-frequency transducer for motility. (The following review «*Sensors* **2014**, *14*, 20825-20842; doi:10.3390/s141120825 » shows the main advances in this direction. Another way is to use a multiplexed 3D ultrasound probe or a sparse array where specific elements can be used to the motility and others for imaging.

The following paragraph has been added in the revised manuscript: Thus, a dual-frequency ultrasound probe can achieve simultaneously propulsion and real time imaging^{38,39}

4. In Figure 2, some modifications are necessary. In Figure 2b, the legend overlays the figure title. In Figure 2d, the title is incomplete. The author should make the necessary modifications to ensure that the figures are clear and easy to understand.

Thank you for bringing this to our attention. We have made the necessary modifications to these figures to ensure they are now clear and easy to understand.